



# Viscosity and phase state of aerosol particles consisting of sucrose mixed with inorganic salts

Young-Chul Song[1,2], Joseph Lilek[3], Jae Bong Lee[4], Man Nin Chan[5,6], Zhijun Wu[7], Andreas Zuend[3],
Mijung Song[1,2,8,*]

[1]Department of Earth and Environmental Sciences, Jeonbuk National University, Republic of Korea

[2]The Earth and Environmental Science System Research Center, Jeonbuk National University, Republic of Korea

[3]Department of Atmospheric and Oceanic Sciences, McGill University, Montréal, Quebec, Canada

[4]Inovative System Safety Research Division, Korea Atomic Energy Research Institute, Republic of Korea

[5]Earth System Science Programme, Faculty of Science, The Chinese University of Hong Kong, Hong Kong, China

[6]The Institute of Environment, Energy, and Sustainability, The Chinese University of Hong Kong, Hong Kong, China

[7]Key Joint Laboratory of Environmental Simulation and Pollution Control, College of Environmental Sciences and Engineering, Peking University, Beijing 100871, China

[8]Department of Environment and Energy, Jeonbuk National University, Republic of Korea

Correspondence: Mijung Song (mijung.song@jbnu.ac.kr)

## Abstract

Research on the viscosity and phase state of aerosol particles is essential because of their significant influence on the particle growth rate, equilibration times and related evolution of mass concentration as well as heterogeneous reactions. So far, most studies of viscosity and phase state have been focused on organic aerosol particles, yet data on how viscosity can vary when the organic materials are mixed with inorganic salts remain scarce. Herein, using a bead-mobility and a poke-and-flow technique, we



quantified viscosities at 293 ± 1 K for binary mixtures of organic material/$H_2O$ and inorganic salts/$H_2O$, as well as ternary mixtures of organic material/inorganic salts/$H_2O$ over the atmospheric relative humidity (RH) range. Sucrose as the organic species, and calcium nitrate ($Ca(NO_3)_2$) or magnesium nitrate ($Mg(NO_3)_2$) as the inorganic salts were examined. For binary sucrose/$H_2O$ particles, the viscosities gradually increased from ~$3 \times 10^{-2}$ to > ~$1 \times 10^8$ Pa s as RH decreased from ~75% to ~25%. Compared with the results for the sucrose/$H_2O$ particles, binary $Ca(NO_3)_2$/$H_2O$ and $Mg(NO_3)_2$/$H_2O$ particles showed drastic enhancements to > ~$1 \times 10^8$ Pa s at low RH close to the efflorescence RH. For ternary mixtures of sucrose/$Ca(NO_3)_2$/$H_2O$ or sucrose/$Mg(NO_3)_2$/$H_2O$, with organic-to-inorganic mass ratios of 1:1, the viscosities of the particles gradually increased from ~$3 \times 10^{-2}$ to greater than ~$1 \times 10^8$ Pa s for RH values from ~75% to ~5%. Compared to the viscosities of the $Ca(NO_3)_2$/$H_2O$ particles, higher viscosities were observed for the ternary sucrose/$Ca(NO_3)_2$/$H_2O$ particles, with values increased by about 1 order of magnitude at 50% RH and about 6 orders of magnitude at 35% RH. Moreover, we applied a thermodynamics-based group-contribution model, AIOMFAC-VISC, to predict aerosol viscosities for the studied systems. The model predictions and viscosity measurements show good agreement within ~ 1 order of magnitude in viscosity. The viscosity measurements indicate that the studied mixed organic–inorganic particles range in phase state from liquid to semi-solid or even solid across the atmospheric RH range at a temperature of 293 K. These results support our understanding that organic/inorganic/$H_2O$ particles can exist in a liquid, semisolid, or even a solid state in the troposphere.

## 1. Introduction

Aerosol particles are emitted from various natural (e.g., the ocean and plants), and anthropogenic (e.g., transportation and fuel combustion) sources, as well as being produced by gas-to-particle conversion and equilibration processes due to chemical processing of gaseous species in the atmosphere (e.g., sulfur dioxide, nitrogen oxides, ammonia, and volatile organic compounds). Submicron-sized aerosol particles mainly consist of organic materials and inorganic salts (Jimenez et al., 2009; Huang et al., 2015; Cheng et al., 2016). Depending on the location and the season, the organic mass fraction of submicron particles ranges from ~20% up to ~90% (Zhang et al., 2007, 2015a; Jimenez et al., 2009; Hao et al., 2014; Huang et al., 2015; Cheng et al., 2016; Wang et al., 2016). Field measurements have shown that organic materials





and inorganic salts are often internally mixed in an aerosol particle (Murphy et al., 2006; Song et al., 2010, 2013). They can significantly affect the local air quality (Kulmala et al., 2011; Wang et al., 2015a; Zhang et al., 2015a; Schmedding et al., 2020), regional climate (Russell et al., 1997; Kaufman et al., 2002; Kanakidou et al., 2005; Kulmala et al., 2011), and human health (Bhattarai et al., 2019). However, the

physicochemical properties of internally mixed particles such as the relative humidity (RH) dependent viscosities and morphologies remain poorly characterized.

The viscosity of aerosols can vary based on the RH and related water uptake, chemical composition, and temperature, with effects on the size distribution (Shiraiwa et al., 2013; Zaveri et al., 2014, 2018), mass concentration of aerosol particles (Shiraiwa and Seinfeld, 2012; Yli-Juuti et al., 2017; Kim et al., 2019),

ice nucleation efficiency (Murray et al., 2010; Ladino et al., 2014; Knopf et al., 2018), and crystallinity of salts (Murray, 2008; Song et al., 2013; Ji et al., 2017; Wang et al., 2017). Depending on the viscosity, the phase state of aerosol particles can be determined; commonly the dynamic viscosity of a liquid is characterized as being less than $10^2$ Pa s, that of a semi-solid is between $10^2$ and $10^{12}$ Pa s, and that of a glassy or crystalline solid is typically greater than $10^{12}$ Pa s (Zobrist et al., 2008; Koop et al., 2011; Reid

et al., 2018).

Most studies on this topic have focused on the determination of the viscosities and phase state of secondary organic aerosols (SOAs) (Mikhailov et al., 2009; Virtanen et al., 2010; Koop et al., 2011; Vaden et al., 2011; Saukko et al., 2012; Renbaum-Wolff et al., 2013; Hao et al., 2014; Kidd et al., 2014; Bateman et al., 2015; Song et al., 2015, 2016a, 2019; Athanasiadis et al., 2016; Grayson et al., 2016a;

Hinks et al., 2016; Hosny et al., 2016; Yli-Juuti et al., 2017; Petters et al., 2019; Wallace and Preston, 2019; Gervasi et al., 2020; Schmedding et al., 2020). Different types of SOAs generated in environmental chambers and flow reactors have been examined. The dynamic viscosities of some types of SOA particles (e.g. generated from toluene and diesel vapors) ranged from $< \sim 10^{-3}$ to $> \sim 10^8$ Pa s, which is approximately equal to the range of viscosities between that of water and tar pitch, depending on the RH, oxidation state,

and chemical composition (Mikhailov et al., 2009; Virtanen et al., 2010; Vaden et al., 2011; Koop et al., 2011; Saukko et al., 2012; Renbaum-Wolff et al., 2013a; Hao et al., 2014; Kidd et al., 2014; Bateman et al., 2015; Zhang et al., 2015b; Athanasiadis et al., 2016; Grayson et al., 2016a,b; Hinks et al., 2016; Hosny et al., 2016; Rothfuss and Petters, 2017; Yli-Juuti et al., 2017; Petters et al., 2019; Song et al., 2015,



2016a,b, 2019,2020; Gervasi et al., 2020). In addition to the viscosity of SOA, the viscosity effects of
inorganic salts should also be investigated to better understand the viscosities and phase states of
atmospherically relevant mixed particles.

Only a few studies have investigated the viscosity and phase state of aerosol particles consisting of organic
materials and inorganic salts. Power et al. (2013) showed using holographic optical tweezers that the
viscosity of sucrose/$H_2O$ particles is significantly greater than that of NaCl/$H_2O$ particles at any RH. They
also showed that the viscosity of mixed sucrose/NaCl/$H_2O$ particles decreased as the NaCl concentration
increased at a certain RH. Rovelli et al. (2019) used a different inorganic salt and observed that the
viscosity of sucrose/$NaNO_3$/$H_2O$ particles decreased as the $NaNO_3$ mass fraction increased at a certain
RH. Richards et al. (2020) reported that inorganic salt in organic and inorganic mixtures increased
viscosity by the ion–molecule effect. Based on field measurements using a particle rebound technique,
Bateman et al. (2017) showed that the phase state of atmospheric submicron aerosol particles consisting
of biogenic organic compounds, sulfate, and black carbon in central Amazonia assumed a nonliquid state
(i.e. viscosity $> 10^2$ Pa s) even at relatively high RH values during periods of anthropogenic influence.
They reported that the aerosol particles influenced by a plume of urban pollution and biomass burning
existed in a non-liquid state (viscosity $> 10^2$ Pa s). Liu et al. (2019) observed atmospheric submicron
aerosol particles in a liquid phase state at RH $> 60\%$ upon the enhancement of the aerosol nitrate fraction
in Shenzhen, which is a subtropical coastal urban city in China. Considering the wide variety of complex
atmospheric aerosol compositions, more information on the viscosity of mixed organic–inorganic
particles is needed.

Using the bead-mobility and poke-and-flow techniques, at $293 \pm 1$ K, we measured the RH-dependent
aerosol viscosities of binary mixtures of organic material/$H_2O$ and inorganic salts/$H_2O$, and ternary
mixtures of organic material/inorganic salts/$H_2O$ under dehydration conditions. Sucrose was selected as
the model organic substance because previous studies have frequently applied it as a surrogate species for
SOA (Zobrist et al., 2011; Power et al., 2013; Grayson et al., 2016b; Song et al., 2016b; Rothfuss and
Petters, 2017; Rovelli et al., 2019). $Ca(NO_3)_2$ and $Mg(NO_3)_2$ were used as model inorganic salts because
they have been commonly observed in mineral dust particles (Usher et al., 2003; Laskin et al., 2005;
Sullivan et al., 2007), and sea-salts (Gupta et al., 2015; Zieger et al., 2017) in the atmosphere (Usher et



al., 2003; Laskin et al., 2005; Sullivan et al., 2007; Shi et al., 2008; Song et al., 2010, 2013; Pan et al., 2017). Moreover, both of these nitrate salts have a relatively low efflorescence RH in aqueous solutions, enabling viscosity measurements of crystal-free solutions from high RH down to at least 30 % RH. Using

these binary and ternary mixtures, we explore how the viscosities vary as a function of RH and associated aerosol compositions. Such viscosity studies can provide a better knowledge of the physicochemical properties of atmospherically relevant aerosol particles consisting of organic material and inorganic salts.

## 2. Experimental

### 2.1 Generation of particles

Sucrose (99.5% purity, Sigma-Aldrich), $Ca(NO_3)_2 \cdot 4\,H_2O$ (99.997% purity, Sigma-Aldrich), and $Mg(NO_3)_2 \cdot 6H_2O$ (99.99% purity, Sigma-Aldrich) were purchased. Solutions containing each compound were prepared at 10 wt. % in pure water (resistivity $\geq$ 18.2 M$\Omega$ cm, Merck Millipore Synergy, Germany). For ternary mixtures, solutions were prepared at an organic-to-inorganic dry mass ratio (OIR) of 1:1. This

dry mass ratio was chosen since it is expected to reveal well the effects of mixing of substantial amounts of both the organic and salt components on overall water uptake and viscosity. For the bead-mobility and poke-and-flow experiments, the aqueous solutions were nebulized on a hydrophobic substrate (Hampton Research, Canada). After nebulizing, the substrate with droplets was set in a flow-cell which was placed below an objective mounted on an optical microscope (Olympus CKX53 with 40× objective) (Pant et al.,

2006; Bertram et al., 2011; Song et al., 2012; Ham et al., 2019). The RH in the cell was adjusted by controlling the $N_2$ mixing ratio of the dry and wet gas flows. The RH in the flow-cell was continuously monitored by a digital humidity sensor (Sensirion, SHT C3, Switzerland). The RH reading from the flow-cell was calibrated by measurement and comparison to the deliquescence RH (DRH) at 293 $\pm$ 1 K of saturated aqueous solutions of ammonium sulfate (80.5%), sodium chloride (76.0%), and potassium

carbonate (44.0%) (Winston and Bates, 1960). The uncertainty of the RH of the humidified $N_2$ flow was $\pm$ 1.5% RH based on the calibration tests. At the initiation of the bead-mobility and poke-and-flow experiments, the RH was maintained at ~85% and the particles equilibrated for ~20 min. Then, the RH was reduced to the target RH (~80 – ~0% RH) and equilibrated for ~30 min for the bead-mobility experiments, and for > ~2 hours for the poke-and-flow experiments. At a given RH, particles that were





20 – 100 μm in diameter were selected for the optical observation. All viscosity measurements were carried out at a temperature of 293 ± 1 K.

## 2.2 Bead-mobility technique

The bead-mobility technique was utilized, which can quantify a range of viscosities from ~$10^{-3}$ to ~$10^3$
Pa s (Renbaum-Wolff et al., 2013b). At the beginning of the bead-mobility experiments, insoluble melamine beads (Cat. no. 86296, Sigma-Aldrich), which were ~1 μm in diameter and dispersed in pure water, were incorporated directly into the droplets deposited on the hydrophobic substrate. Aerosol particles were then placed into the flow-cell, as described in Sect. 2.1. The total gas flow in the cell was 1200 sccm for the bead-mobility experiments. The continuous gas flow generates a shear stress on a
particle's surface, which yields internal circulations of the beads in the particles during measurements. The movement of the beads at 293 ± 1 K was recorded every 1 s with a CMOS camera (MSC-M 3.0 UCMOS, China) and then quantified at a target RH. The viscosity was calculated from the bead speeds using a calibration line, which produced bead speeds for the sucrose particles at different RH values (see Fig. S1 of the Supplement). Figure S2 illustrates the mean bead speeds of each system as a function of
the RH. When the bead speeds within a particle are too slow to be observed with this technique (i.e. below ~$10^{-6}$ µm·m s$^{-1}$ corresponding to ~$10^2$ Pa s, see Fig. S1 and S2), we used the poke-and-flow technique (Sect. 2.3). For example, the movement of the beads within $Mg(NO_3)_2/H_2O$ particles at ~35% RH was too slow to be readily observed (Fig. S2). Further information on the calibration and the bead-mobility technique is given in Sect. S1 in the Supplement.


## 2.3 Poke-and-flow technique

Murray et al. (2012) developed the poke-and-flow technique for phase state determination, which was expanded for the quantification of viscosities > ~$10^2$ Pa s by Renbaum-Wolff et al. (2013a), Grayson et al. (2015), and Song et al. (2015). This technique uses a small hole on the top of the flow-cell to allow us
to poke the particles on the hydrophobic substrate using a sterilized sharp needle (Jung Rim Medical Industrial Co., South Korea). Using a micromanipulator (Narishige, model MO-152, Japan), the needle was controlled in the *x*-, *y*-, *z*-direction. The needle was passed through from the top to the bottom of a





particle and then the needle was removed. After poking, the geometrical change of a deposited particle was observed and recorded by an optical microscope (Olympus CKX53 with 40× objective) with a CCD

camera (MSC-M 3.0 UCMOS, China). All experiments were carried out at 293 ± 1 K.

Figure S3 shows an example of the geometrical changes in the sucrose/$H_2O$, sucrose/$Ca(NO_3)_2$/$H_2O$, and sucrose/$Mg(NO_3)_2$/$H_2O$ particles during pre-poking, poking, and post-poking. Before poking by the needle, the particles had a geometry of a spherical cap. After poking by a needle, a half-torus geometry with an inner hole in the particle was observed. As time progressed, the particle recovered by adopting

its original geometry by filling of the hole to minimize the surface energy. The experimental flow time ($\tau_{(exp, flow)}$) was obtained when the area of the inner hole just after poking ($t = 0$ s. Fig. S3) decreased to 1/4 of the initial inner hole area (Renbaum-Wolff et al., 2013a; Grayson et al., 2015, 2016; Song et al., 2019). Figure S4 shows the $\tau_{(exp, flow)}$ at different RH for sucrose/$H_2O$, sucrose/$Mg(NO_3)_2$/$H_2O$, and sucrose/$Ca(NO_3)_2$/$H_2O$ particles. The $\tau_{(exp, flow)}$ of the particles was converted to the lower limit to the

viscosity using the equation reported in Sellier et al. (2015). Using the poke-and-flow technique, the upper limit to viscosity is unknown in this work. For binary mixtures of inorganic salts/$H_2O$, we were unable to determine the viscosity between $10^2$ to $10^8$ Pa s because the poke-and-flow technique is not accessible for droplets that are supersaturated with respect to a crystalline state of involved inorganic salts. We also measured the RH value and corresponding viscosity at which the particles cracked. When the particles

cracked upon poking, without any detectable flow after a duration of > 5 h at a given RH, we defined a lower limit to the viscosity of $1 \times 10^8$ Pa s based on the results of Renbaum-Wolff et al. (2013a), Grayson et al. (2015, 2016), and Song et al. (2019). Further details on these experiments can be found in Sect. S2 of the Supplement.

**2.4 AIOMFAC-VISC model**

The Aerosol Inorganic–Organic Mixtures Functional groups Activity Coefficients Viscosity model (AIOMFAC-VISC) is a thermodynamics-based group-contribution model for predicting the viscosity of aqueous organic mixtures (Gervasi et al., 2020). It is an extension module to the AIOMFAC model, which explicitly describes interactions among organic functional groups, inorganic ions, and water (Zuend et al.,

2008, 2011). The AIOMFAC-VISC model offers predictive estimates of mixture viscosity covering the





range from liquid-like, low-viscosity aqueous solutions to highly viscous, organic-rich amorphous solutions in the atmospherically relevant temperature range. An online version of AIOMFAC (AIOMFAC-web), which includes viscosity predictions for aqueous organic mixtures, can be run at https://aiomfac.lab.mcgill.ca.

Aside from the focus on the viscosity effects of organic compounds, in recent work AIOMFAC-VISC has been further developed to enable the prediction of the viscosity of aqueous electrolyte solutions and aqueous inorganic–organic mixtures; with details provided elsewhere (Lilek et al., *in preparation*). Briefly, a semi-empirical approach was introduced to predict the mixture viscosity of aqueous electrolyte solutions as a function of temperature, ion activities, and ionic strength. Model parameters for the aqueous

ion interaction approach were simultaneously fitted to room temperature viscosity measurements aggregated by Laliberté for many electrolyte solutions (Laliberté, 2007, 2009). The training dataset for AIOMFAC-VISC did not include measurements for the nitrate salts investigated in this study. For aqueous electrolyte solutions, AIOMFAC-VISC predictions currently achieve an excellent level of accuracy, comparable to that of the fitted expressions by Laliberté (2007). Furthermore, to more

accurately capture the water uptake behavior of sucrose as a function of equilibrium RH (i.e. water activity of the solution), the version of AIOMFAC-VISC used to produce the viscosity predictions for the sucrose-containing systems in this work includes an improved treatment of the ether-group–water interactions of sucrose. Thus, viscosity predictions in this study for aqueous sucrose differ slightly from those of the AIOMFAC-web model version.

For the viscosities of ternary mixtures, a Zdanovskii-Stokes-Robinson (ZSR) type mixing rule is applied to predict the viscosity of the ternary mixtures of organic material/inorganic salts/$H_2O$. This mixing rule is here adopted for viscosity applications since the AIOMFAC-VISC model does presently not include explicit ion–organic interaction effects on viscosity (only on activity coefficients). Therefore, a mixing rule is required to combine the predictions of the viscosity contributions from the electrolyte-free

subsystem (sucrose/$H_2O$) and those from the organic-free aqueous electrolyte subsystem for the viscosity estimation of the full mixture. Generally, the ZSR approach involves combining some physical property of two or more (binary) mixtures at the same RH, often to determine the water content of the whole multicomponent mixture (Zdanovskii, 1936, 1948; Stokes and Robinson, 1966). Following the





experimental conditions, the shown AIOMFAC-VISC predictions for the ternary systems use an OIR of

1:1, which constrains at each RH level the fractional contributions of mass ($f_1$, $f_2$) from each of the subsystems, sucrose/$H_2O$ (1) or salt/$H_2O$ (2). The viscosity of the overall mixture is then obtained as $\ln(\eta/\eta°) = f_1 \ln(\eta_1/\eta°) + f_2 \ln(\eta_2/\eta°)$, with $\eta°$ denoting unit viscosity (1 Pa s); see details in section S5 of the Supplement.

**3. Results and discussion**

**3.1 Viscosities of particles consisting of sucrose/$H_2O$**

Figure 1 shows the RH-dependent viscosities of sucrose/$H_2O$ particles obtained using the bead-mobility and poke-and-flow techniques. The viscosities of sucrose/$H_2O$ particles were determined to be between $\sim 2 \times 10^{-1}$ and $\sim 1 \times 10^{1}$ Pa s for RH values of ~85– 69%, and between $\sim 5 \times 10^{3}$ and $\sim 2 \times 10^{6}$ Pa s for RH

values of ~50 – 37% (Fig. 1). The particles containing sucrose/$H_2O$ cracked at ~23% RH when poked with a needle, and restorative flow did not occur over a time span of 5 h (Fig. 2a). Consequently, only a lower limit for the viscosity of $\sim 10^{8}$ Pa s was obtained (Renbaum-Wolff et al., 2013a; Grayson et al., 2015, 2016; Song et al., 2015a, 2019). Figure 1 also includes results from previous studies for the viscosities of sucrose/$H_2O$ particles using different techniques such as bead-mobility, poke-and-flow,

holographic optical tweezers, and fluorescence lifetime imaging microscopy (Hosny et al., 2013, Power et al. 2013, Grayson et al., 2015, Song et al., 2015, 2016b). As shown in Fig. 1, the viscosities for the sucrose/$H_2O$ particles from this study and previous studies are consistent within ~ 1 order of magnitude at given RH values. Using the entire dataset, it suggests that the sucrose/$H_2O$ particle are in a liquid phase state for RH > ~65%, in a semi-solid phase state for ~25% < RH < ~65%, and in a semi-solid or solid

phase state for RH < ~23%.

**3.2 Viscosities of particles consisting of Mg(NO₃)₂/$H_2O$ or Ca(NO₃)₂/$H_2O$**

Despite the abundant mass of inorganic species in the atmosphere (Jimenez et al., 2009; Cheng et al., 2016), few studies have reported the viscosity of single particles consisting of inorganic species covering

the RH range to high salt concentrations (Power et al., 2013; Rovelli et al., 2019). Herein, the RH-



dependent viscosity was quantified for particles consisting of $Mg(NO_3)_2/H_2O$ or $Ca(NO_3)_2/H_2O$ determined at $293 \pm 1$ K upon dehydration using the bead-mobility and poke-and-flow techniques.

Figure 3 illustrates the RH-dependent viscosities of $Mg(NO_3)_2/H_2O$ or $Ca(NO_3)_2/H_2O$ particles upon dehydration. The obtained viscosities of the $Mg(NO_3)_2/H_2O$ particles varied from ~$6 \times 10^{-3}$ to ~$4 \times 10^{-2}$
Pa s for RH values from ~70 to ~35% whereas the values for $Ca(NO_3)_2/H_2O$ ranged from ~$3 \times 10^{-2}$ to ~$9 \times 10^{1}$ Pa s for RH values from ~65 to ~10%. Previous studies have reported the viscosities of $Ca(NO_3)_2/H_2O$ and $Mg(NO_3)_2/H_2O$ solutions only at high RH values based on bulk solution measurements (Fig. 3) (Abdulagatov et al., 2004; Wahab et al., 2006). The values of the viscosities of the particles at high RH are consistent with our results within ~ 1 order magnitude despite a limited number
of data points due to the solubility limit restricting bulk solution measurements to the RH range above 70 %. In addition, Fig. 3 shows the viscosities of $NaNO_3/H_2O$ particles measured by other groups (Haynes, 2015; Rovelli et al., 2019). The viscosities of the $NaNO_3/H_2O$ particles were similar to those of the $Ca(NO_3)_2/H_2O$ particles for the RH range from ~30 to ~100 %.

During the poke-and-flow experiments, the $Mg(NO_3)_2/H_2O$ or $Ca(NO_3)_2/H_2O$ particles cracked when
poked with a needle at RH values of 30% and 5%, respectively. At these RH levels, noticeable restorative flow did not occur for over 5 h (Fig. 2b and c), which resulted in a lower estimated limit for the viscosity (~$10^8$ Pa s) at the given RH values. The RH value where the particles shattered is similar to the efflorescence RH (ERH) of $Mg(NO_3)_2/H_2O$ and, in case of $Ca(NO_3)_2$. The ERH of $Mg(NO_3)_2/H_2O$ is known to be ~30% at 298 K (Li et al., 2008; Wang et al., 2015b) and RH of crystallization for
$Ca(NO_3)_2/H_2O$ is reported to be ~7% at 298 K (Liu et al., 2008). In this study, we optically observed an ERH of $32.0 \pm 2.5$% for $Mg(NO_3)_2/H_2O$ particles (i.e. Fig. S5b), but we did not optically observe the ERH of $Ca(NO_3)_2/H_2O$ at $293 \pm 1$ K with decreasing RH (Fig. S5c). Liu et al. (2008) also observed optically no efflorescence point for $Ca(NO_3)_2/H_2O$ particles down to 0% RH using an optical microscope, but they confirmed the crystallization for $Ca(NO_3)_2/H_2O$ particles at RH of ~7% by Raman spectroscopy
at 298 K. The RH values at which shattering of particles occurred for the binary inorganic salt particles of $Mg(NO_3)_2/H_2O$ or $Ca(NO_3)_2/H_2O$ were close to their reported ERH values and/or the RH of crystallization. When comparing the two inorganic salts, $Ca(NO_3)_2/H_2O$ particles showed slightly higher viscosities than the $Mg(NO_3)_2/H_2O$ particles, i.e. approximately 1 order of magnitude higher at equivalent





RH for the RH range from ~60 to 30% (Fig. 3). The difference in viscosities is likely due to the higher

hygroscopicity of dissolved $Mg(NO_3)_2$ compared to $Ca(NO_3)_2$ (Guo et al., 2019). The $Ca(NO_3)_2/H_2O$ particles were of a liquid phase state at RH > ~10%, and a semi-solid or solid phase state at RH < ~5%. The $Mg(NO_3)_2/H_2O$ particles were of a liquid phase state at RH > ~35%, and a semi-solid or solid phases state at RH < ~30%. Based on the viscosity measurement, both inorganic particles underwent a phase change from liquid (<$10^2$ Pa s) to semi-solid or solid (>$10^2$ Pa s ) within a narrow RH range (Fig. 3)

compared to the sucrose/$H_2O$ particles (Fig. 1). Indeed, this discontinuity in viscosity with decreasing RH suggests a phase transition.

### 3.3 Viscosities of particles consisting of sucrose/$Ca(NO_3)_2/H_2O$ or sucrose/$Mg(NO_3)_2/H_2O$

To explore the viscosity of more atmospherically relevant aerosol particle configurations, we investigated

ternary systems containing sucrose mixed with either $Ca(NO_3)_2$ or $Mg(NO_3)_2$ for OIR of 1:1 at 293 ± 1 K. Shown in Fig. 4 are the viscosities for the sucrose/$Ca(NO_3)_2/H_2O$ (Fig. 4a) and sucrose/$Mg(NO_3)_2/H_2O$ particles (Fig. 4c) upon dehydration using the bead-mobility and the poke-and-flow measurement techniques.

The viscosities of the sucrose/$Ca(NO_3)_2/H_2O$ particles ranged from ~$3 \times 10^1$ to ~$3 \times 10^{-2}$ Pa s at ~45 <

RH < ~80%, and from ~$9 \times 10^5$ to ~$3 \times 10^4$ Pa s at ~25 < RH < ~30% (Fig. 4a). At ~13% RH, when poked with a needle, the sucrose/$Ca(NO_3)_2/H_2O$ particles cracked. Thereafter, the particles did not exhibit any notable flow behavior for over 5 h (Fig. 5a), producing a lower limit for the viscosity of ~$10^8$ Pa s. This result indicates that the sucrose/$Ca(NO_3)_2/H_2O$ particles were in a liquid phase state at RH > ~ 45%, in a semi-solid state at ~25% < RH < ~30%, and in a semi-solid or solid state for RH < ~13%.

For sucrose/$Mg(NO_3)_2/H_2O$ particles, the viscosities ranged from ~$7 \times 10^{-2}$ to ~$1 \times 10^1$ Pa s for RH values ranging from ~70 to ~35%, and from ~$2 \times 10^4$ to ~$2 \times 10^5$ Pa s for the RH ranging from ~17 to ~11% (Fig. 4c). The sucrose/$Ca(NO_3)_2/H_2O$ particles cracked at ~6% RH without flow over 5 h (Fig. 5b); and thus the lower limit of the viscosity was determined to be ~$10^8$ Pa s. These results imply that the sucrose/$Ca(NO_3)_2/H_2O$ particles are in a liquid phase state at RH > ~ 35%, a semi-solid phase state for

17% < RH < 11%, and a semi-solid or solid phase state at RH < ~6%. As shown in Figs. 4a and 4c, the viscosities of the sucrose/$Mg(NO_3)_2/H_2O$ particles are approximately 1 order of magnitude lower at 50%





RH, and ~6 orders of magnitude lower at 35% RH than the sucrose/Ca(NO$_3$)$_2$/H$_2$O particles over the same RH range. Both particles experienced a phase state change from a liquid phase state to a semi-solid or even solid phase state with decreasing RH. Finally, although not confirmed by our measurements, it is possible that in one or both of these ternary mixtures a gel phase transition may occur upon sufficient dehydration, as has been observed in mixtures of gluconic acid with CaCl$_2$ (Richards et al., 2020).

### 3.4 Model–measurement comparison of viscosity

Figure 4b and 4d shows the model–measurement comparison of the RH-dependent viscosities of the binary and ternary systems. For the viscosities of sucrose/H$_2$O, the viscosities from the AIOMFAC-VISC model prediction agreed with the measurements within ~ 1 order of magnitude. For the viscosities of Ca(NO$_3$)$_2$/H$_2$O and Mg(NO$_3$)$_2$/H$_2$O, the viscosity prediction from the AIOMFAC-VISC model showed a good agreement until the measured inorganic salt viscosities change steeply, indicating crystallization or another phase transition that the model ignores. When comparing AIOMFAC-VISC viscosity predictions to the measurements in this study, note that AIOMFAC-VISC assumes that mixtures remain in a metastable state to low water activity (or high solution concentration) and that solutes do not crystallize, which may explain the discrepancy among the model predictions and the measurements in the low RH region shown in Fig. 4b and 4d. Moreover, we note that the training of the AIOMFAC-VISC electrolyte solution model parameters did not include data from this study. The model predictions at RH levels below ~20% represent extrapolations of the model beyond the range of experimental data used in its training.

In AIOMFAC-VISC, the prediction of the temperature-dependent pure-component viscosity is based on the experimentally determined, yet uncertain glass transition RH. As such, we include error thresholds of ± 5% in the pure-component glass transition RH in Fig. 4 (Gervasi et al., 2020). Such uncertainty estimates are not necessary for aqueous electrolyte mixtures, so the AIOMFAC-VISC model sensitivity is shown instead, which is defined as the viscosity change due to a ± 2% change in the mass fraction of water of the solution, while the ratio of the other components is preserved (Gervasi et al., 2020). The uncertainty in the aqueous sucrose viscosity dominates the modelled uncertainty of viscosity estimates for the ternary mixtures. For the viscosities of ternary sucrose/Ca(NO$_3$)$_2$/H$_2$O or sucrose/Mg(NO$_3$)$_2$/H$_2$O solutions, AIOMFAC-VISC viscosity predictions show agreement with the measurements within about



1 order of magnitude from high RH to about 30 % RH (considering measurement uncertainty). In the case
of the ternary sucrose/Ca(NO$_3$)$_2$/H$_2$O system, the model–measurement deviation increases to about 1.5
orders of magnitude in viscosity at 30% RH and lower, with AIOMFAC-VISC underestimating the
measured viscosity. This result may be explained at least in part by the model predicting substantially
lower viscosities for the binary aqueous Ca(NO$_3$)$_2$ system in this lower RH range, which affects the
predictions for the ternary system via the deployed mixing rule. The good agreement between model and
measurements for the ternary sucrose/Mg(NO$_3$)$_2$/H$_2$O system, even at low RH levels, may be interpreted
as indicative of suppressed salt crystallization in the presence of sucrose, since no discontinuities in
viscosity are observed (in contrast to the measurements for the binary salt particles).

**4. Conclusion and atmospheric implications**

Herein, we measured the RH-dependent viscosities at 293 $\pm$ 1 K for particles consisting of organic
material/H$_2$O, inorganic salt/H$_2$O, and organic material/inorganic salt/H$_2$O upon dehydration using the
bead-mobility and poke-and-flow techniques. We selected sucrose as the organic species because
previous studies have frequently applied it as a surrogate species of SOA and this organic offers favorable
properties for measurements. Ca(NO$_3$)$_2$ and Mg(NO$_3$)$_2$ were selected as the inorganic salts for viscosity
measurements because these inorganic salts have been frequently observed from mineral dust and sea salt
particles (Usher et al., 2003; Laskin et al., 2005; Sullivan et al., 2007). For the binary mixtures, the
obtained viscosity of the sucrose/H$_2$O particles agreed well with those reported in previous studies, i.e.
mixture viscosities $<10^2$ Pa s at RH $>$ ~65%, which corresponds to a liquid phase state; mixture viscosities
of ~$10^2$ to $10^8$ Pa s at RH values between ~65 and ~25%, which correspond to a semi-solid phase state;
and mixture viscosities $>$ ~$10^8$ Pa s at RH $<$ ~25%, which correspond to semi-solid or amorphous (or
crystalline) solid phase states (Power et al., 2013; Grayson et al., 2016b; Song et al., 2016b; Rothfuss and
Petters, 2017; Rovelli et al., 2019). Upon dehydration, we also quantified the viscosities of the inorganic
salts. The viscosities of the Mg(NO$_3$)$_2$/H$_2$O particles were to be $<$ ~$4 \times 10^{-2}$ Pa s for RH $>$ ~35%, and $>$
~$10^8$ Pa s for RH $<$ ~31%, whereas those of the Ca(NO$_3$)$_2$/H$_2$O particles were $<$ ~$9 \times 10^0$ Pa s for RH $>$
~10% and $>$ ~$10^8$ Pa s for RH $<$ ~10%. The particles containing either of these two inorganic salts cracked
upon poking when the RH reached a value near the salt's ERH and/or the phase transition RH from a





droplet to a solid phase state. These inorganic particles exhibited a sudden enhancement in the viscosity when the particle effloresced. In contrast, sucrose/$H_2O$ particles showed a smooth enhancement in the

viscosity with decreasing RH; this means that the viscosity of sucrose/$H_2O$ particles gradually approached their glass transition RH. The AIOMFAC-VISC model prediction and viscosity measurements showed a good agreement within ~ 1 order of magnitude, especially at RH levels above 30%, where applicable. For sucrose/$Ca(NO_3)_2$/$H_2O$ particles and sucrose/$Mg(NO_3)_2$/$H_2O$, predicted and measured viscosities showed good agreement over the whole RH range.

The phase states of aerosol particles have an impact on rate and potential for heterogeneous reactions as well as consequences for the resulting mass concentration of aerosol particles. The uptake coefficient of gas phase oxidants depends on the phase states of aerosol particles (George et al., 2010; Xiao et al., 2011; Kuwata et al., 2012; Slade and Knopf, 2014; Davies and Wilson, 2015;  Berkemeier et al., 2016; Li et al., 2020; Xu et al., 2020). For example, ozone uptake coefficients ($\gamma O_3$) decreased by an order of magnitude

in a semi-solid phase state compared to a liquid phase state (Steimer et al., 2015). Moreover, the effective mass concentration of aerosol particles can depend on the (assumed or modelled) phase state of aerosol particles (Shiraiwa and Seinfeld, 2012; Yli-Juuti et al., 2017; Kim et al., 2019). If a liquid aerosol phase state is assumed, the mass concentration may be overpredicted by up to one order of magnitude (Shiraiwa and Seinfeld, 2012). Based on the measurements and calculations, our results show that the studied

aerosol particles consisting of organic material/inorganic salt/$H_2O$ range from liquid to semi-solid or solid phase states depending on the RH. A caveat is that a single OIR of 1:1 and relatively simple aerosol systems were used compared to the multicomponent-multiphase particles likely occurring in the real atmosphere (Murphy et al., 2006; Jimenez et al., 2009; Song et al., 2010, 2013; Huang et al., 2015; Cheng et al., 2016). Additional investigations are required to further explore and quantify how the viscosities

and phase states of mixed organic–inorganic particles vary with the OIR, temperature, and functional group complexity.



*Data availability*. Underlying material and related data for this paper are provided in the Supplement.

*Author contributions*. MS and YS designed this study. MS and JBL setup and calibrated the viscosity instrument. YS and MS conducted viscosity experiments and analyzed the data. JL and AZ conducted AIOMFAC-VISC model predictions. YS and MS prepared the manuscript with contributions from JL, JBL, AZ, ZJ, and MN.

*Competing interests*. The authors declare that they have no conflict of interest.

*Acknowledgements*. This work was supported by the National Research Foundation of Korea (NRF) grant funded by the Korea government (MSIT) (NRF-2019R1A2C1086187), and by the Fine Particle Research Initiative in East Asia Considering National Differences (FRIEND) Project (2020M3G1A1114548). This
project was undertaken with the financial support of the government of Canada through the federal Department of Environment and Climate Change (grant no. GCXE20S049). This work was also supported by Alfred P. Sloan Foundation under Prime Award no. G-2020-13912 and the Regents of the University of California. M. Song gives thanks to D. Ham for the technical support.





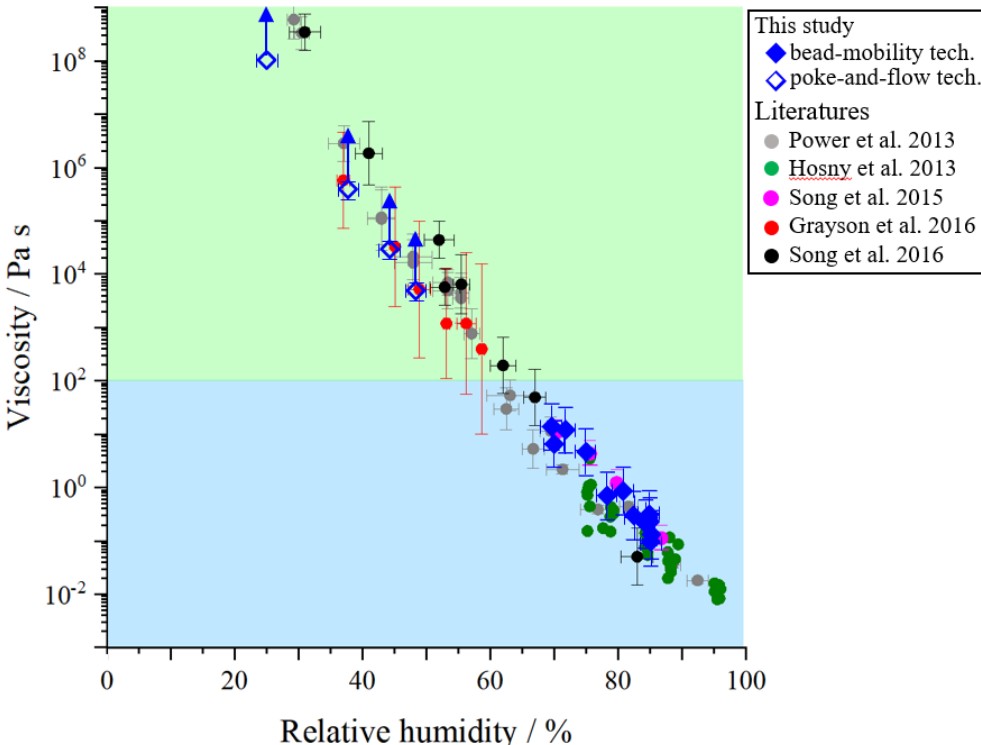


**Figure 1.** Measured dynamic viscosities of sucrose/$H_2O$ particles compared with previous studies at 293 K (Hosny et al., 2013, Power et al. 2013, Grayson et al., 2015, Song et al., 2015, 2016b) and this study. The error bars in relative humidity (RH) were generated from the calibrated RH uncertainty and the standard deviation from grouped viscosity data (grouped by RH ranges) which included at least 3 data points. The error bars in viscosities were produced by 95% prediction bands of viscosities (Fig. S1). Upward arrows represent lower limit to the viscosities of the particles that calculated by the experimental flow time and the equation reported in Sellier et al. (2015). Details are described in Fig. S2. Light blue and green regions indicate liquid and semi-solid regions, respectively.





| RH | Pre-poking | Poking | After 0 second | After 2 hours | After 5 hours |
|---|---|---|---|---|---|
| (a) Sucrose /$H_2O$ 24% RH | | | | | |
| (b) $Mg(NO_3)_2$ /$H_2O$ 30% RH | | needle | | | |
| (c) $Ca(NO_3)_2$ /$H_2O$ 6% RH | | | | | |

Figure 2. Optical images during poke-and-flow experiments at the points of pre-poking, poking, and post-poking for particles consisting of (a) sucrose/$H_2O$, (b) $Mg(NO_3)_2$/$H_2O$, and (c) $Ca(NO_3)_2$/$H_2O$ for relative humidity (RH) where the particles shattered. The scale bar is 5 µm.

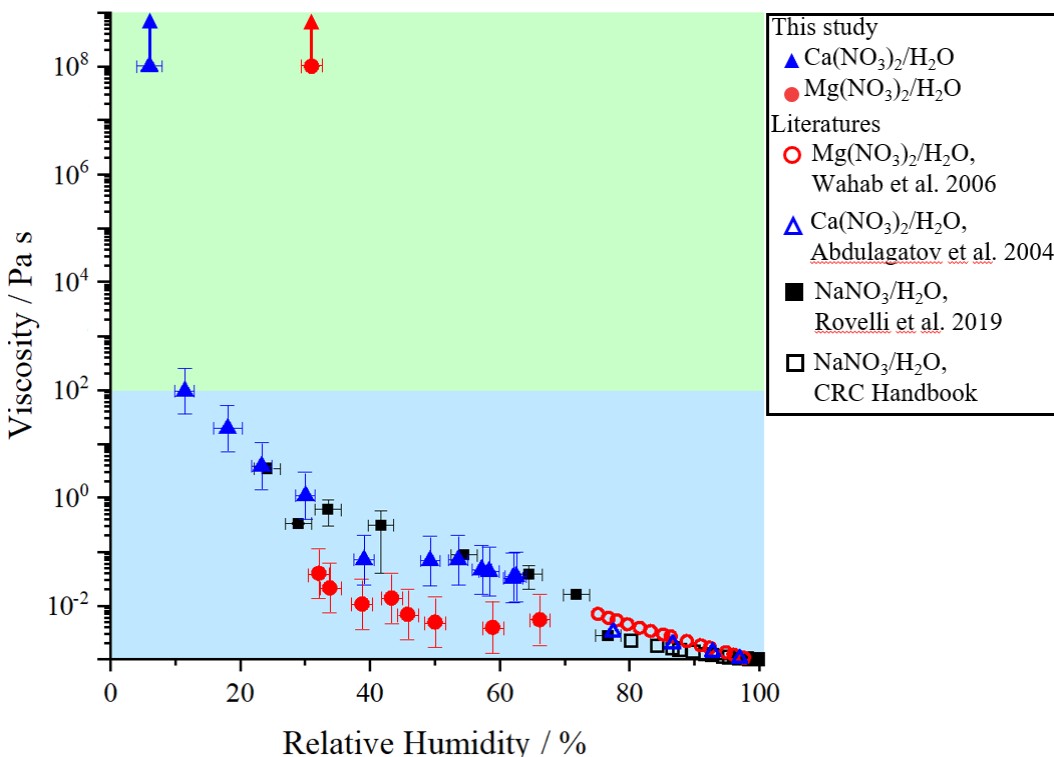


**Figure 3**. Viscosities of inorganic salts containing $Ca(NO_3)_2/H_2O$ and $Mg(NO_3)_2/H_2O$, and previous results (Abdulagatov et al., 2004; Wahab et al., 2006; Haynes, 2015) and this study. Also, viscosities of $NaNO_3/H_2O$ from previous studies are included (CRC Handbook; Rovelli et al., 2019). Upward arrows represent a lower limit for the viscosities of the particles calculated by the experimental flow time and the

equation reported in Sellier et al. (2015). The error bars in relative humidity (RH) were generated from combined RH sensor uncertainty and the standard deviation from the grouped viscosity data by RH, which included at least 3 data points. The error bars in viscosities were produced by 95% prediction bands of viscosities (Fig. S1). Light blue region indicates a liquid phase, and light green region indicates a semi-solid phase.





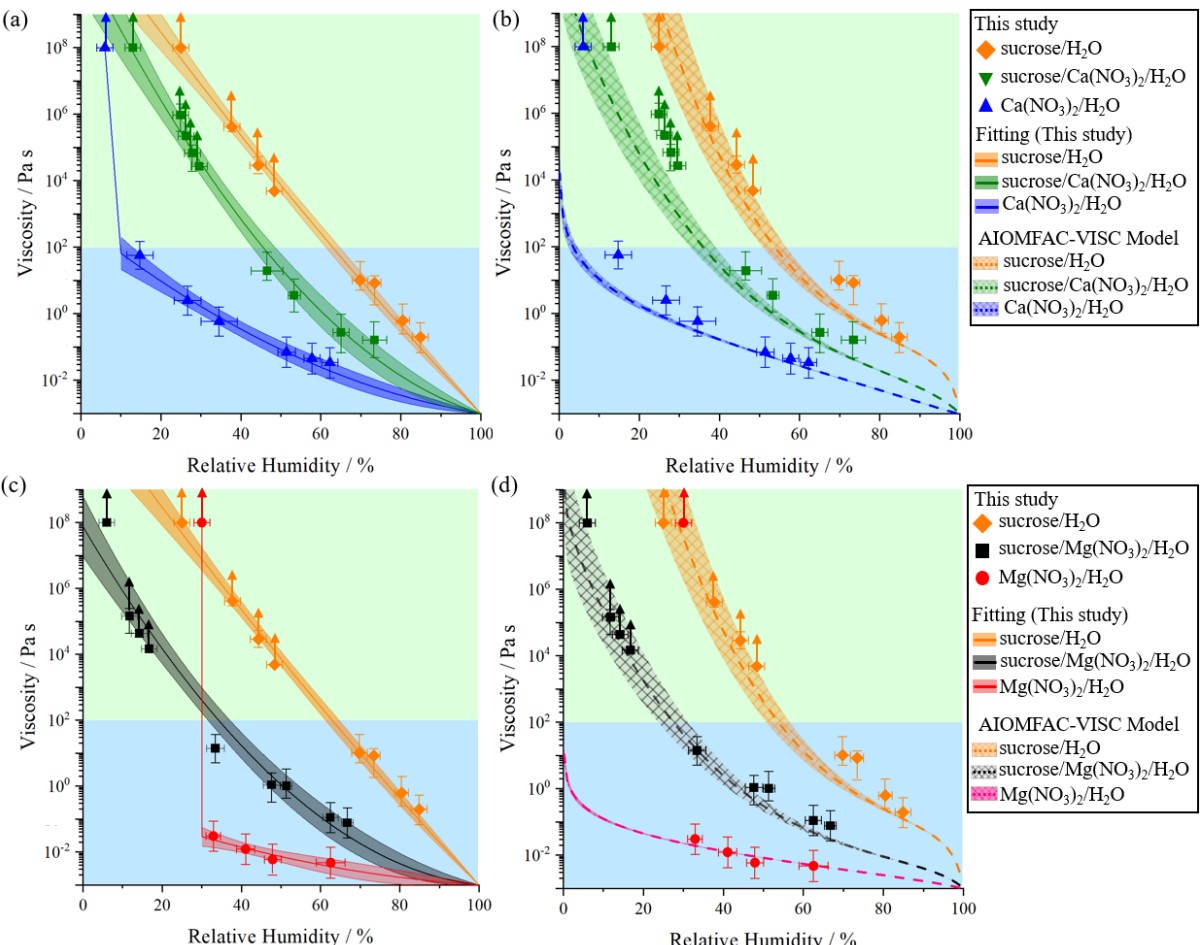


**Figure 4**. Comparison of viscosities of binary and ternary systems obtained from the bead-mobility and poke-and-flow experiments. (a) Mean viscosities of sucrose/$H_2O$, $Ca(NO_3)_2$/$H_2O$, and sucrose/$Ca(NO_3)_2$/$H_2O$ particles with polynomial fits from measured viscosities for an organic-to-inorganic mass ratio (OIR) of 1:1. All data at a temperature of 293 ± 1 K , (b) Mean viscosities of

sucrose/$H_2O$, $Ca(NO_3)_2$/$H_2O$ (grouped by relative humidity (RH) using data from Fig. 3), and sucrose/$Ca(NO_3)_2$/$H_2O$ particles with AIOMFAC-VISC model predictions, (c) mean viscosities of sucrose/$H_2O$, $Mg(NO_3)_2$/$H_2O$ (grouped by RH using data from Fig. 3), and sucrose/$Mg(NO_3)_2$/$H_2O$ particles with polynomial fits  from measured viscosities for an OIR of 1:1, (d) mean viscosities of sucrose/$H_2O$, $Mg(NO_3)_2$/$H_2O$, and sucrose/$Mg(NO_3)_2$/$H_2O$ particles with AIOMFAC-VISC model

predictions. The error bars in RH were generated from the RH uncertainty and the standard deviation of individual data grouped by RH. The error bars in viscosity represent grouped viscosity by RH including





at least 3 data points. Symbols marked with upward arrows represent lower limits of the viscosities of the

particles calculated based on the experimental flow time period and the equation reported in Sellier et al.

(2015). Fitting (solid line curves and shaded areas) from measured viscosities are based on 2[nd] polynomial

curve fits with 95% confidence bands. Predictions (dash line curves with checker shaded areas) are based

on from AIOMFAC-VISC model. The parameterizations are described in Sect. S3 and Table S1. Light

blue and green regions represent liquid and semi-solid phase state ranges, respectively.

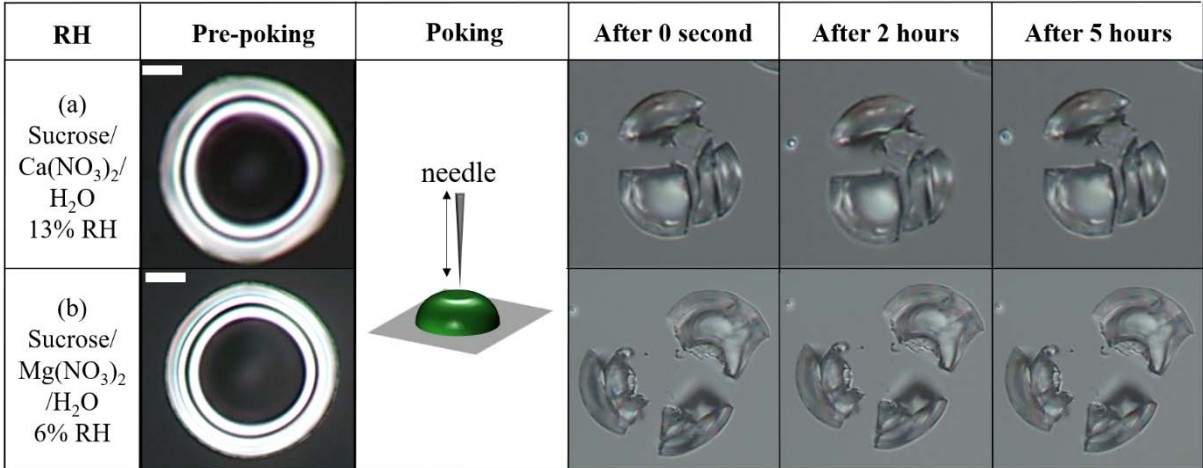


Figure 5. Optical images during poke-and-flow experiments at the points of pre-poking, poking, and post-poking for particles consisting of (a) sucrose/Ca(NO$_3$)$_2$/H$_2$O particles and (b) sucrose/Mg(NO$_3$)$_2$/H$_2$O particles for relative humidity (RH) where the particles shattered. The scale bar indicates 5 µm.





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
