# Peer review of "Viscosity and phase state of aerosol particles consisting of sucrose mixed with inorganic salts"

_Atmospheric Chemistry and Physics, 2021_

## Author Comment (AC1)

We thank the referees for carefully reading our manuscript and for their valuable comments. Listed below are our responses in blue to the comments from the referees of our manuscript.

**Response to Referee #1**

Summary: The manuscript describes very nice and comprehensive measurement results of viscosity for binary and ternary mixtures of sucrose, calcium/magnesium nitrate, and water, using the bead-mobility and poke-and-flow techniques that cover two ranges of viscosity. A thermodynamic model was also used to obtain viscosity of those mixtures and to compare with those from measurements. The extension of viscosity measurements to more complex aerosol composition is surely important to understand the physical properties of atmospheric particles. The study is also well designed, and results well interpreted. I therefore recommend Minor Revision, with some minor comments below.

General Comments of Referee #1

*[1]* It is interesting to compare results in Figure 4b and 4d. When AIOMFAC-VISC predicted better for the inorganic salt (magnesium nitrate), it also predicted better for the corresponding ternary mixture in Figure 4d. Does this result mean that ZSR-type mixing rule suffices for viscosity prediction? That is, if one can capture the viscosity of the individual component well, one can predict the viscosity of mixed components well, at least for non-reactive and non-interacting mixtures such as those in this study? I am also particularly interested in why the mixing rule works on a natural-log basis (P9/L227). Any physical reason behind that?

*[A1]* Thank you for your comment. We agree with this observation. AIOMFAC-VISC predicts the viscosity for aqueous magnesium nitrate better than that of aqueous calcium nitrate, which is the primary reason for the better agreement observed with the ternary sucrose-Mg(NO$_3$)$_2$ mixture. With a better prediction for aqueous calcium nitrate at lower water content, we would expect closer agreement with the measurements for the ternary sucrose-Ca(NO$_3$)$_2$ mixture. One reason for this difference is that at low RH (high solute concentration), the Ca(NO$_3$)$_2$ measurements in this study do not agree very well with the experimental training data used to fit the model parameters, whereas the Mg(NO$_3$)$_2$ measurements are more consistent with the

training data.

We expect that a ZSR-type mixing rule suffices for non-reactive and non-interacting mixtures that exist as Newtonian fluids over a wide RH range. Some aqueous electrolytes, especially those with divalent cations, have been observed to undergo a gel transition at low RH (Cai et al., 2015; Richards et al., 2020a). Such gel phase transitions are not explicitly accounted for by AIOMFAC-VISC, which may pose a challenge for the ZSR mixing rule. However, to answer that question more quantitatively, experimental data covering a wider range of electrolytes and organic compounds in mixed organic–inorganic solutions will be necessary. Such data are scarce.

There is a mathematical and a physical reason for the log-basis applied in our version of a ZSR mixing rule for viscosity. The natural-log basis relates to the range of observed viscosities, which spans many orders of magnitude. For example, a 1:1 mixture of water ($10^{-3}$ Pa s) and maple syrup ($\sim 1$ Pa s) will have a viscosity with an order of magnitude roughly halfway between those of the two component liquids. A log-based "linear" mixing rule accomplishes this, while a non-log-based mixing rule based on an arithmetic mean of viscosities will likely overpredict the mixture viscosity, being biased toward the liquid with the higher single-liquid viscosity. A log-based mixing rule for viscosity was first expressed by Arrhenius as the mole-fraction-weighted sum of natural logarithms of the viscosities of component liquids. Furthermore, consider that the exponential of an arithmetic mean of log(x) values is mathematically equivalent to the corresponding geometric mean of the non-log values, e.g., $\exp[(\ln(0.001) + \ln(1.0))/2] = \mathrm{sqrt}[0.001 * 1.0]$, and both means can be appropriately weighted (e.g., by mole fractions). Given the physical finding that the viscosity of a liquid solution can be approximately modeled by an exponential Arrhenius-type equation involving the activation energy for viscous flow, it is reasonable to apply a (weighted) geometric mean of viscosities in a mixing rule.

References:

Cai, C., Tan, S., Chen, H., Ma, J., Wang, Y., P. Reid, J., and Zhang, Y.: Slow water transport in $MgSO_4$ aerosol droplets at gel-formingrelative humidities, Phys. Chem. Chem. Phys., 17, 29 753–29763, https://doi.org/10.1039/C5CP05181A, 2015.

Richards, D. S., Trobaugh, K. L., Hajek-Herrera, J., Price, C. L., Sheldon, C. S., Davies, J. F.,

Davis, R. D.: Ion-molecule interactions enable unexpected phase transitions in organic-inorganic aerosol. Sci. Adv., 6 (47), 1–12. https://doi.org/10.1126/sciadv.abb5643, 2020a.

*[2]* P5/L138: Any reason for using two different equilibrating durations for these two techniques? And indication of water evaporation is complete for 30 min equilibrating time, especially for more viscous particles?

[*A2*] Particles were conditioned to the surrounding relative humidity for ~30 min prior to the bead-mobility experiment and > 2 h prior to the poke-and-flow experiment. Based on viscosity ranges and the corresponding conditioning times as described in previous studies (Grayson et al., 2015; Song et al., 2019; McClean et al., 2021), we also used different experimental conditioning times for equilibrium with the gas-phase water vapor prior to the experiments. With the bead-mobility technique, which measured viscosities of less than ~ $10^2$ Pa s, and the size range studied in this work (20 ~ 100 μm in diameter), the mixing times of water within the sucrose particle can be calculated in less than 10 min. With the poke-and-flow technique, which determined the lower limit of the viscosities between ~$10^4$ and ~$10^6$ Pa s, and the size range studied (20 ~ 50 μm in diameter), the mixing times of water within the sucrose particle can be calculated in less than 2 h. Therefore, the different conditioning times were used in the experiments and the times would be sufficient for near equilibrium conditions with the gas-phase water. To clarify, the following revised text will be added to Sect 2.1:

"The RH was reduced to the target RH, and particles then conditioned to the surrounding RH for ~30 min for the bead-mobility experiments, and for > ~2 hours for the poke-and-flow experiments to give sufficient time for reaching equilibrium with the surrounding water vapor."

References:

Grayson, J. W., Song, M., Sellier, M. and Bertram, A. K.: Validation of the poke-flow technique combined with simulations of fluid flow for determining viscosities in samples with small volumes and high viscosities, Atmos. Meas. Tech., 8(6), 2463–2472, doi:10.5194/amt-8-2463-2015, 2015.

Maclean, A. M., Smith, N. R., Li, Y., Huang, Y., Hettiyadura, A. P. S., Crescenzo, G. V., Shiraiwa, M., Laskin, A., Nizkorodov, S. A. and Bertram, A. K.: Humidity-Dependent Viscosity of Secondary Organic Aerosol from Ozonolysis of β-Caryophyllene: Measurements, Predictions, and Implications, ACS Earth Sp. Chem., 5(2), 305–318, doi:10.1021/acsearthspacechem.0c00296, 2021.

Song, M., Maclean, A. M., Huang, Y., Smith, N. R., Blair, S. L., Laskin, J., Laskin, A., DeRieux, W.-S. W., Li, Y., Shiraiwa, M., Nizkorodov, S. A. and Bertram, A. K.: Liquid-liquid phase separation and viscosity within secondary organic aerosol generated from diesel fuel vapors, Atmos. Chem. Phys., 19(19), 12515–12529, doi:10.5194/acp-2019-367, 2019.

*[3]* P12/L309: Any specific evidence to suggest that at least one of these mixtures went through a gel transition? Are the electrolytes in the study (calcium, magnesium, and nitrate) capable of forming contact ion pairs as magnesium-sulfate pair and calcium-gluconate pair? Would love to see the viscosity measurement results of magnesium sulfate, which has been suggested to form gel.

*[A3]* Thank you for your comment. Unfortunately, we do not have any evidence of a gel transition occurring in one of our mixture systems. However, a few papers have shown that aerosol particles consisting of calcium nitrate or magnesium sulfate undergo a gel transition of aqueous electrolytes at low RH (Cai et al., 2015; Richards et al., 2020a,b). It is possible that in the mixture particles containing $Ca^{2+}$, and $Mg^{2+}$ cations, a gel phase transition may occur upon sufficient dehydration, as has been observed in mixtures of gluconic acid with $CaCl_2$ (Richards et al., 2020a). We will cite both papers in Sect. 3.3.

References:

Cai, C., Tan, S., Chen, H., Ma, J., Wang, Y., P. Reid, J., and Zhang, Y.: Slow water transport in MgSO4 aerosol droplets at gel-formingrelative humidities, Phys. Chem. Chem. Phys., 17, 29 753–29763, https://doi.org/10.1039/C5CP05181A, 2015.

Richards, D. S., Trobaugh, K. L., Hajek-Herrera, J., Price, C. L., Sheldon, C. S., Davies, J. F., Davis, R. D.: Ion-molecule interactions enable unexpected phase transitions in organic-inorganic aerosol. Sci. Adv., 6 (47), 1–12. https://doi.org/10.1126/sciadv.abb5643, 2020a.

Richards, D. S., Trobaugh, K. L., Hajek-Herrera, J. and Davis, R. D.: Dual-Balance Electrodynamic Trap as a Microanalytical Tool for Identifying Gel Transitions and Viscous Properties of Levitated Aerosol Particles, Anal. Chem., 92(4), 3086–3094, doi:10.1021/acs.analchem.9b04487, 2020b.

*[4]* It would be nice to see the results being put in a broader context of viscosity measurements using other techniques (e.g., particle rebound and particle merging etc.) for similar species, if any.

*[A4]* Thank you for your comment. To address the Referee's comment, we will state different techniques including fluorescent lifetime imaging and aerosol optical tweezers from previous studies in Fig. 1 and Fig. 3.

*[5]* P13/L359: please delete "to be" after "were".

*[A5]* Thank you for the correction. We will correct this wording.

---

## Author Comment (AC2)

We thank the referees for carefully reading our manuscript and for their valuable comments. Listed below are our responses in blue to the comments from the referees of our manuscript.

**Response to Referee #2**

Summary: In this paper the authors quantify the change in viscosity of particles comprised of inorganic salts mixed with sucrose. I think the paper is well written and there is a clear discussion around the experimental procedure and results shown. It is important that more studies quantify the behaviour of mixed inorganic-organic systems and it is refreshing to see a submission focusing on targeted laboratory studies rather than broad extrapolation to global conditions. Therefore, I believe the article can be published in ACP subject to a few general points raised below.

General Comments of Referee #2

*[1]* Whilst I do fully support the focus on simple mixtures, it would be nice for the authors to provide more context on why these salts were chosen and the expected source of particles with these mass ratios studied. Apologies if I missed this in the document.

*[A1]* Thank you for the comment. In Sect. 1, we have described the reason why we selected the salts and organic compound as the model compounds as below:

*"Sucrose was selected as the model organic substance because previous studies have frequently applied it as a surrogate species for SOA (Zobrist et al., 2011; Power et al., 2013; Grayson et al., 2016b; Song et al., 2016b; Rothfuss and Petters, 2017; Rovelli et al., 2019). Ca(NO3)2 and Mg(NO3)2 were used as model inorganic salts because they have been commonly observed in mineral dust particles (Usher et al., 2003; Laskin et al., 2005; Sullivan et al., 2007), and sea-salts (Gupta et al., 2015; Zieger et al., 2017) in the atmosphere (Usher et al., 2003; Laskin et al., 2005; Sullivan et al., 2007; Shi et al., 2008; Song et al., 2010, 2013; Pan et al., 2017). Moreover, both of these nitrate salts have a relatively low efflorescence RH in aqueous solutions, enabling viscosity measurements of crystal-free solutions from high RH down to at least 30 % RH."*

*[2]* Likewise, can the authors comment on the expected timescales for equilibration under ambient conditions? This does not detract from the important of providing more data on previously unstudied systems.

*[A2]* Based on the work by Koop (2011) and Shiraiwa et al. (2011), the expected timescales for equilibration of various particles ranged from seconds to years depending on the viscosities and materials. Ambient aerosol particles may equilibrate with the gas-phase water vapor within a few seconds to hours or even years. However, further studies are needed to confirm the equilibrating timescales of mixed organic–inorganic particles under ambient conditions, including the timescale for the equilibration of semivolatile organics and inorganics (aside from that of water). Added complexity in particle morphology, such as phase separation, can also influence the equilibration timescale (e.g. Huang et al. 2021); however, our systems did not exhibit phase separation for the studied mixing ratios. To address the Referee's comment, the following text will be added to the manuscript in Sect. 4.

"Further studies are needed to confirm the equilibrating timescales of mixed organic–inorganic particles under ambient conditions, including the timescale for the equilibration of semivolatile organics and inorganics. Added complexity in particle morphology, such as phase separation, can also influence the equilibration timescale (e.g. Huang et al. 2021); however, our systems did not exhibit phase separation for the studied mixing ratios."

References:

Huang, Y., Mahrt, F., Xu, S., Shiraiwa, M., Zuend, A. and Bertram, A. K.: Coexistence of three liquid phases in individual atmospheric aerosol particles, Proc. Natl. Acad. Sci. U. S. A., 118(16), e2102512118, doi:https://doi.org/10.1073/pnas.2102512118, 2021.

Koop, T., Bookhold, J., Shiraiwa, M., Pöschl, U. and Poeschl, U.: Glass transition and phase state of organic compounds: dependency on molecular properties and implications for secondary organic aerosols in the atmosphere, Phys. Chem. Chem. Phys., 13(43), 19238–55, doi:10.1039/c1cp22617g, 2011.

Shiraiwa, M., Ammann, M., Koop, T. and Pöschl, U.: Gas uptake and chemical aging of semisolid organic aerosol particles., Proc. Natl. Acad. Sci. U. S. A., 108(27), 11003–8, doi:10.1073/pnas.1103045108, 2011.

*[3]* Minor comment Page 8, line 224: 'the shown AIOMFAC-VISC predictions for the ternary systems use an OIR of..' please change this to 'the AIOMFAC-VISC..'

*[A3]* Thank you for the comment. We will revise the manuscript accordingly.